# Preoperative Magnetic Resonance Image and Computerized Tomography Findings Predictive of Facial Nerve Invasion in Patients with Parotid Cancer without Preoperative Facial Weakness—A Retrospective Observational Study

**DOI:** 10.3390/cancers14041086

**Published:** 2022-02-21

**Authors:** Won Ki Cho, Min Kyoung Lee, Young Jun Choi, Yoon Se Lee, Seung-Ho Choi, Soon Yuhl Nam, Sang Yoon Kim

**Affiliations:** 1Department of Otolaryngology—Head and Neck Surgery, Asan Medical Center, University of Ulsan, College of Medicine, Seoul 05505, Korea; onekey91@naver.com (W.K.C.); shchoi@amc.seoul.kr (S.-H.C.); synam2@amc.seoul.kr (S.Y.N.); sykim2@amc.seoul.kr (S.Y.K.); 2Department of Radiology, Asan Medical Center, University of Ulsan, College of Medicine, Seoul 05505, Korea; tosky333mk@gmail.com (M.K.L.); jehee23@gmail.com (Y.J.C.)

**Keywords:** parotid cancer, facial nerve, invasion, MRI, CT, prognosis, margin, size

## Abstract

**Simple Summary:**

Facial nerve invasion in parotid cancer affects survival outcomes as well as functional outcomes after surgery-based treatment. Normal facial muscle function before surgery does not always exclude the possibility of involvement of the facial nerve by a tumor. Especially in patients without facial palsy, accurate evaluation of invasion before surgery is necessary to plan optimal facial nerve resection and reconstruction. Various findings are obtained from preoperative radiological findings, such as CT and MRI. We evaluated the role of these radiological findings in predicting nerve invasion. Large tumor, spiculated margin, and anterolateral location may suggest a high risk of nerve involvement even in patients with normal preoperative facial function. These findings may help surgeons to avoid unexpected facial nerve invasion and to make adequate surgical plans to get optimal oncological and functional outcomes.

**Abstract:**

(1) Background: Facial nerve resection with reconstruction helps achieve optimal outcomes in the treatment of facial nerve invasion (FNI) of parotid cancer. Preoperative imaging is crucial to predict facial nerve reconstruction. The radiological findings of CT or MRI may predict FNI in the parotid cancer even without facial paralysis. Methods: We retrospectively reviewed the records of 151 patients without facial nerve paralysis before surgery who had undergone tumor resection. Previously untreated parotid cancers were included. (2) Results: The median follow-up duration was 62 months (range: 24–120 months). The FNI (+) group (*n* = 30) showed a significantly worse 5-year overall survival compared with the FNI (−) group (75.5 vs. 93.9%; hazard ratio = 4.19; 95% confidence interval: 1.74–10.08; *p* = 0.001). The tumor margin, tumor size, presence in the anterolateral parotid region (area 3), retromandibular vein involvement, distance from the stylomastoid foramen to the upper tumor margin, and a high tumor grade were significant factors related to FNI in the univariate analysis. A spiculated tumor margin, the tumor size (2.2 cm), and presence in area 3 were factors predicting FNI in the logistic regression model (*p* = 0.020, 0.005, and 0.050, respectively; odds ratio: 4.02, 6.40, and 8.16, respectively). (3) Conclusions: The tumor size (≥2.2 cm), spiculated margin, and presence in area 3 as presented in CT and MRI may help clinicians preoperatively predict FNI in patients with parotid cancer and establish an appropriate surgical plan.

## 1. Introduction

Malignant parotid tumor accounts for 20% to 30% of parotid gland tumors [1]. Surgical resection is a mainstay in the treatment of parotid cancer, and the extent of surgery is determined by prognostic factors indicating aggressive tumor behaviors [2]. Tumor stage, histology, grade, lymph node metastasis, resection margin, and facial nerve invasion (FNI) are prognostic factors indicative of wide excision with adjuvant therapy. Among them, clinical and pathological FNI causes a poor prognosis [3,4]. 

FNI is observed in the following histological types of parotid cancer: adenoid cystic carcinoma (50%), adenocarcinoma not otherwise specified (42%), squamous cell carcinoma (22%), and mucoepidermoid carcinoma (20%) [5]. High-grade parotid cancer invades surrounding tissue more frequently, including the facial nerve, than low- or intermediated-grade parotid cancer. The grade of the tumor is difficult to confirm before surgery. Clear resection margin is also crucial for successful local control. Therefore, careful evaluation of FNI is required to secure a safe margin and to improve treatment outcomes before surgical resection of parotid cancer.

Upon confirming FNI, the involved portion of the facial nerve should be resected, and its marginal status should be confirmed by onsite biopsy. Resection of the involved facial nerve results in facial paralysis, which deteriorates the patient’s quality of life [6]. Immediate facial nerve repair shows better results than delayed reconstruction. The treatment outcome of facial nerve grafting and the nerve substitution method for facial reanimation have the disadvantages of scar formation and aberrant sprouting of the nerve ends induced by delayed reconstruction after parotid surgery [7]. 

Therefore, before surgery for parotid cancer, the risk associated with facial nerve resection owing to tumor infiltration into the facial nerve should be evaluated, and a surgeon should be consulted regarding the reconstruction strategy. Preoperative facial weakness, pain, high tumor grade, and advanced tumor stage are risk factors for FNI in parotid cancer [8], although they may not always lead to FNI. Tumors located in the superficial or deep parotid gland lobe have been identified using various radiological examinations. Ultrasonography (US), computerized tomography (CT), and magnetic resonance image (MRI) are commonly used radiological modalities to evaluate parotid tumors. 

US is often utilized to guide fine-needle aspiration cytology and visualize tumor characteristics rather than identify the location or extent of parotid tumors. MRI better demonstrates the soft tissue structure of the parotid gland and surrounding structures than CT, with which the extratemporal portion of the facial nerve cannot be traced directly [9]. The stylomastoid foramen (SMF), retromandibular vein, and Utrecht line are used to define the depth of parotid tumors related to the facial nerve [10,11]. A multiplanar analysis of these landmarks with the parapharyngeal space is likely to be superior to the analysis of a single landmark in evaluating tumor location and establishing the surgical plan.

In addition to tumor location, radiological findings predicting FNI help prepare for facial nerve reanimation surgery and secure an adequate surgical margin. Most MRI and CT findings used to evaluate FNI depend on the radiologists’ or clinicians’ experience [12]. More reliable findings suggesting FNI in parotid cancer are required for preparing an adequate surgical plan. Therefore, this study aimed to evaluate the predictive efficacy of radiological findings suggesting FNI in parotid cancer.

## 2. Materials and Methods

### 2.1. Study Patients

We conducted a retrospective observational study by reviewing the electronic medical records of patients diagnosed with parotid malignancy at our tertiary referral hospital from January 2006 to December 2016 (Figure 1). The inclusion criteria were age ranging from 18 to 80 years and previously untreated, biopsy-proven parotid carcinoma after operation. The enrolled patients underwent ultrasonography-guided fine-needle aspiration biopsy, followed by contrast-enhanced CT of the head and neck, with or without MRI. 

The patients underwent thorough a physical examination for facial nerve palsy, followed by parotidectomy with or without neck dissection. Total or radical parotidectomy was performed with resecting the involved facial nerve. Partial or superficial parotidectomy was applied for accidental parotid cancer (proven cancer after operation), confined lesion, and low-grade tumor. Exclusion criteria were previous treatments, the initial presentation of inoperable advanced diseases, preoperative facial palsy, distant metastasis, and loss to follow-up within 2 years after treatment. Patients with an interval of more than 3 months from the imaging workups to the operation were also excluded because of the possibility of discrepancy from the initial radiological findings. 

All records had detailed descriptions of FNI available in the operation notes, and the presence of FNI was evaluated on the basis of the operation findings retrospectively. FNI was defined as gross tumor infiltration confirmed by operators, which was also confirmed by pathologists. The subsequent sacrifice of the involved facial nerve was performed to achieve a tumor-negative margin. The histological grade was assessed by a pathologist according to the 2017 World Health Organization classification [13]. This study was reviewed and approved by the institutional review board of our hospital (2019-0976), and the requirement of obtaining informed consent was waived.

### 2.2. Variables

#### 2.2.1. Radiological Variables

The following variables on CT and/or MRI of patients with parotid cancer were reviewed independently by two radiologists who had experience of 12 years (Y. J. C.) and 19 years (J. H. L): tumor size, margin (well-defined, spiculated, and ill-defined), SMF involvement, status of the retromandibular vein (intact, obliterated, and displaced), distance from the SMF to the upper tumor margin, tumor location in the course of the main facial nerve trunk, presence in the anterolateral parotid region, and the radiologists’ overall impression of facial nerve involvement. 

The margins were described as well-circumscribed, indistinct (ill-defined), or spiculated (characterized by lines radiating from the mass). FNI was suspected by consensus between the two radiologists. Based on previous radiological studies, artificial boundaries were drawn using anatomical structures within the parotid gland as reference, and a newly made area in each boundary of the parotid gland was expected to include the facial nerve trunk or division. 

On axial CT and/or MRI, the parotid gland was divided into areas 1, 2, and 3 or the medial, posterolateral, and anterolateral regions, respectively. An imaginary line from the SMF to the retromandibular vein was drawn to define area 1. Subsequently, an additional line was projected laterally from a point 0.5 cm behind the retromandibular vein to divide the superficial parotid gland into areas 2 and 3 (Figure 2A). Previous studies reported that the facial nerve trunk within 8–20 mm from the SMF divided it into the temporofacial and cervicofacial branches (Figure 2B). Tumors located within 20 mm from the SMF on sagittal or coronal CT and/or MRI and in the overlapping portion of areas 1 and 2 on axial images were considered to be located in the course of the main facial nerve trunk. 

Conversely, tumors located more than 20 mm from the SMF on sagittal or coronal CT and/or MRI and/or in area 3 on axial images were considered to be located in the peripheral nerve branch. In addition to objective measures, two experienced radiologists reviewed CT and/or MRI scans and determined the subjective impression of FNI blindly. The histological FNI status was also blinded for radiologists. The area classification was validated by two head and neck surgeons again (Y. S. L. and S. C). Using these parameters, significant radiological and clinicopathological factors were analyzed.

#### 2.2.2. Clinical and Pathological Variables

The following clinical data were included as variables: age (>60 years), gender, smoking history (>10 pack-years), alcohol intake (≥1 drink/day), and Charlson comorbidity index (≥1). The following pathological data were included as variables: tumor size (mm), histological grade (G1/G2/G3), extracapsular spread, lymphovascular invasion, perineural invasion, involvement of resection margin, and pathological tumor and nodal staging.

### 2.3. Magnetic Resonance Imaging

Magnetic resonance imaging was obtained on a 1.5-T MR scanner (Achieva, Philips Medical Systems, Best, The Netherlands or Intera, Philips Medical System) or on a 3-TMR scanner (Achieva, Philips Medical Systems) using a 16-channel neurovascular coil (Sense NV coil, Philips Medical Systems). MR examinations were performed with a 3-T MR imaging unit (Achieva, Philips Medical Systems) with a 16-channel neurovascular coil (SENSE NV coil, Philips Medical Systems). 

A transverse T2-weighted turbo spin-echo sequence was performed with a repetition time msec/echo time msec of 3906/100, with two signals acquired and an acquisition time of 2 min 59 s. A transverse T1-weighted turbo spin-echo sequence was performed with 675/10, two signals acquired, and an acquisition time of 2 min 51 s. All T1- and T2-weighted images were acquired with 30 imaging sections, a field of view of 190 (anterior to posterior) × 190 (right to left) × 120 (feet to head) mm^3^, and a reconstruction voxel size of 0.37 × 0.37 × 3.00 mm^3^. An intravenous dose of 0.1 mmol per kilogram of body weight of gadoterate meglumine (Dotarem; Guerbet, Paris, France) was administered to all patients to obtain contrast-enhanced T1-weighted images.

### 2.4. Computerized Tomography

Contrast-enhanced CT using either a Somatom Sensation 16 (Siemens Medical Solutions, Erlangen, Germany) or LightSpeed QX/I multidetector CT scanner (GE Medical System, Milwaukee, WI, USA) was used. The scanning parameters included the following: 3 mm slice thickness, 20.9 cm field of view (FOV), 120 kV of voltage, 200 mA of current, and a 256 × 256 matrix. A 140 mL of intravenous bolus dose of nonionic iodinated contrast agent (iopromide; Ultravist 300; Bayer Schering Pharma AG, Berlin, Germany) was injected. Continuous scans with 3 mm collimation were obtained at 3 mm intervals with no gap from the skull base to the upper chest for axial images and from the mandible posterior to the pharynx for coronal images.

### 2.5. Statistical Analysis

Statistical analyses were performed with IBM^®^ SPSS^®^ Statistics version 24.0 for Windows (IBM Corp., Armonk, NY, USA) and MedCalc version 19.1.7 (MedCalc Software, Ostend, Belgium). The 5-year overall and recurrence-free survival rate of enrolled patients was compared using Kaplan–Meier analysis and logrank test. The *t*-test for continuous variables and the χ^2^ or Fisher’s exact test for categorical variables were performed to analyze significant factors affecting FNI, assuming a multivariate normal distribution followed by individual planned hypothesis testing. The cutoff tumor size and distance from the SMF to the upper tumor margin on CT and/or MRI were determined with the ROC curve analysis and AUC estimation. 

Univariate and multivariate regression models were used to identify significant predictors of FNI. Variables with *p*-value < 0.05 in univariate analyses were selected for multivariate regression analyses with the backward elimination method. The hazard ratio (HR) and 95% confidence interval (CI) were estimated. Variables in univariate analysis were chosen to estimate AUC and diagnostic values with ROC curve analysis. The performance of the model was subsequently compared with the overall radiologists’ impression by comparing the AUC estimates of all ROC curves. All statistical tests were two-tailed, and a *p*-value < 0.05 was considered statistically significant.

## 3. Results

### 3.1. Characteristics of Patients with or without FNI

This study enrolled 151 patients, including 73 (48.3%) men and 78 (51.7%) women, with a median age of 55 years (range: 19–80 years). Preoperative CT were evaluated for all patients, while MRI was used for 127 patients (84.1%). The median follow-up duration was 62 months (range: 24–120 months). The most common pathologic type of parotid carcinoma was mucoepidermoid carcinoma (*n* = 48, 29.6%), irrespective of the presence or absence of FNI. 

In patients without FNI, mucoepidermoid carcinoma (*n* = 38, 31.4%) was followed by carcinoma ex pleomorphic adenoma (*n* = 20, 16.5%). In patients with FNI, mucoepidermoid carcinoma (*n* = 10, 33.3%) was followed by adenoid cystic carcinoma, salivary ductal carcinoma, and carcinoma ex pleomorphic adenoma (*n* = 6, 20% for all). FNI was identified in 30 (19.9%) patients. Table 1 summarizes the clinical and pathological data of the FNI (+) and (−) groups. All clinical demographics, except for gender, were comparable between patients with and without FNI. 

The mean pathological tumor size was larger in the FNI (+) group than in the FNI (−) group (30.6 vs. 24.1 mm, *p* < 0.001). A high tumor grade, an advanced T/N category, perineural invasion, extracapsular extension, and involvement of resection margin were risk factors for FNI, while lymphovascular invasion presented with a tendency for FNI (*p* = 0.096). Radical parotidectomy was performed in all patients with FNI, and a total parotidectomy and partial or superficial parotidectomy were performed in 75 (62.0%) and 46 (38.0%) patients. The proportions of patients who underwent adjuvant radiation therapy or concurrent chemoradiation therapy after surgical resection were higher in the FNI (+) group (90%) than in the FNI (−) group (44%) (*p* < 0.001). 

At the last follow-up, 126 (83.4%) patients were alive without disease; 14 (9.3%) patients had died of the disease; six (4.0%) patients had died of other causes; and four (2.6%) patients were alive with disease. The 5-year overall survival rate of all patients was 90.0% (95% CI: 84.7–95.3%). This was worse among patients with FNI (75.5%) than among those without FNI (93.9%; HR = 4.19; 95% CI: 1.74–10.08; *p* = 0.001). The 5-year recurrence-free survival rate of all patients was 82.4% (75.9–88.9%) and was not significantly different between patients with and without FNI (85.4 vs. 69.8%; HR = 2.21; 95% CI: 0.98–4.96; *p* = 0.056; Figure 3).

### 3.2. Clinicopathological and Radiological Factors Predicting FNI

Clinicopathological and radiological factors predicting FNI were analyzed separately in patients with and without FNI based on operative findings of the facial nerve (Table 2). Among the radiological factors, univariate analyses revealed that spiculated margin, radiological tumor size ≥ 2.2 cm, presence in area 3 (i.e., in the course of the facial nerve divisions), and retromandibular vein involvement (obliteration) were significantly associated with FNI (*p* = 0.003, 0.001, 0.014, and 0.023, respectively). ROC curve analyses showed that the cutoff tumor size predicting FNI was 2.2 cm (*p* < 0.001, Figure 4A) and that the cutoff distance from the SMF to the upper tumor margin was 3 mm (*p* = 0.07, Figure 4B). 

The distance from the SMF to the upper tumor margin presented marginal significance for predicting FNI (*p* = 0.060, Table 2). The imaging impression described by two radiologists was a significant factor predicting FNI (*p* < 0.001). Among 30 patients with FNI, 21 (70.0%) had been suspected with FNI based on imaging findings by two radiologists. Among 121 (80.1%) patients without FNI, 38 (31.4%) had been suspected with FNI based on preoperative CT and/or MRI imaging. 

In multivariate analyses using significant objective parameters, spiculated margin (*p* = 0.016), radiological tumor size ≥ 2.2 cm (*p* = 0.007), and presence in area 3 (*p* = 0.042) increased the risk for FNI by approximately four, six, and nine times, respectively (all *p <* 0.05, Table 3). In terms of the histological grade, high-grade carcinoma increased the risk for FNI by approximately 3.7-fold more compared to low- or intermediate-grade carcinoma (95% CI: 0.86–5.85; *p* < 0.017). Regarding the invasion site in the overall course of the facial nerve, the site of division (*n* = 24, 80.0%) was most commonly invaded, followed by the facial nerve trunk (*n* = 4, 13.3%) and both sites (*n* = 2, 6.7%).

We explored the diagnostic performances of these radiological risk factors, including the tumor size, speculated margin, and presence in area 3, using the ROC curves for discriminating FNI. We added the histological grade to the radiological risk factors to evaluate the diagnostic value, and its AUC was 0.839. The difference in AUC between the four predictive variables and the imaging impression was statistically significant at 0.146 (95% CI: 0.023–0.228; *p* = 0.016, Figure 4C).

The AUC of the three predictive variables and radiologists’ impressions were 0.787 and 0.693, respectively, with no statistically significant difference *(p* = 0.129; Table 4, Figure 4D). Based on the optimal cutoff values of the predictors in the ROC curve analysis, the diagnostic performance was estimated. Three radiological factors with or without the histological grade presented with a better diagnostic accuracy compared to the radiologists’ subjective impression (Table 5).

## 4. Discussion

Perineural spread of parotid cancers is a negative prognostic factor and an indication of adjuvant radiation therapy. Clinicopathological factors, such as an advanced tumor stage (T3 or T4), advanced nodal stage (N2 or N3), and a high histological grade, increase the risk of parotid cancer induced FNI. In addition to these aggressive tumor characteristics, the location and marginal status of parotid cancer in the parotid parenchyma based on radiological findings predicted FNI. 

In this study, radiological factors, including spiculated tumor margin, large tumor size, and presence in area 3, were associated with FNI and presented better diagnostic accuracy than the subjective impression described by radiologists. Preoperative facial weakness, high tumor grade, and large tumor size are risk factors for post-parotidectomy facial nerve palsy [14,15]. The latter two factors are mainly intraoperative or pathological findings, which are not beneficial to prepare for facial reanimation before surgery. 

In addition to preoperative facial weakness, other preoperative factors predictive of FNI have not been well-elucidated. A lower response to preoperative electroneuronography could imply FNI and poor postoperative facial nerve function [16]. However, this electrophysiological study for parotid cancer without definite facial weakness is not routinely used. Instead, routine preoperative radiological modalities, such as CT and MRI, are utilized to determine the anatomical relationship between the tumor and the surrounding structures, particularly the invisible facial nerve, and surgical extent. 

The impression of radiologists is based on the tumor morphology (spiculated margin or obscure margin), tumor size, and invasion to adjacent structures, suggesting facial nerve invasion. The criteria of the apparent diffusion coefficient (ADC) and tumor location are variable between radiologists. Tumor histology and grade were not considered. Tumor location, retromandibular vein invasion, and tumor grade were added to these factors in this study. Therefore, we intended to find parameters predictive of FNI using these radiological workups. 

Previous studies mainly focused on the tumor location rather than tracing the course of the extratemporal facial nerve, which separates the superficial and deep lobes of the parotid gland. Although MRI is the best modality for visualizing the soft tissue structure, the extratemporal portion of the facial nerve cannot be traced directly [9]. The tumor location should be evaluated precisely because deep-lobe tumors are predisposed to causing postoperative facial weakness and suggest FNI in cases of cancer [15]. MRI tractography has recently been developed to detect the perineural spread of parotid cancers [17]. 

Direct visualization of the facial nerve using the three-dimensional double-echo steady state with water excitation sequence and cross-sectional CT and MRI demonstrated a high NPV for FNI [18,19]. These techniques require specific modalities, which are not widely used. In contrast, the indirect methods using the facial nerve line, retromandibular vein, and Utrecht line are affected by the tumor size, tumor location, and specific methods described previously [20,21]. The retromandibular vein presented with high diagnostic accuracy (63.5% to 86.4%) and specificity (85.7% to 96.2%) along with an inconsistent sensitivity (29.6% to 71%) [10,22]. 

These studies did not factor in the radiologists’ experience with FNI. We intended to develop objective parameters using the more reliable anatomical landmarks that can be visualized on CT or MRI. The bony structures and retromandibular vein are relatively easy to identify on CT and MRI. This study presented that three radiological parameters had a 77% sensitivity, 73% specificity, and 93% NPV for FNI, which were consistent and comparable to previous studies. Among 206 patients who underwent parotid surgery for parotid cancers, 37 patients were excluded due to lack of CT or MRI within 90 days before surgery. Exclusion of these patients may be another selection bias for this study.

Considering tumor growth and morphological change during more than 90 days, however, the closer the time interval between the imaging studies and the operation is, the more accurate the imaging studies would be. If we enrolled all patients regardless of the time interval between imaging studies and surgery, underestimated patients would be included in this study, which could be another selection bias. Comparison between the included and the excluded patients according to the time interval between the imaging studies and the operation would be another future study to guide an exact or acceptable time point to achieve successful parotid surgery.

A parotid tumor size of ≥5 cm measured intraoperatively was reported to abut the facial nerve [23]. A large tumor is presumed to be close to the facial nerve, considering the limited space in the parotid parenchyma. Similarly, presence in area 3 implies that the tumor is located in the distal branches of the intraparotid facial nerve. The lack of parotid parenchyma between the facial nerve and the tumor facilitates tumor invasion into the nerve. 

The spiculated margin is a sonographic finding of thyroid nodules, which are strongly suggestive of cancer, and a CT or MRI finding of parotid tumors. The histological grade showed a higher diagnostic accuracy compared to the radiologists’ impression. However, preoperative US-guided fine- or core-needle aspiration biopsy suggested limited findings of histological grade, which were confirmed using complete tumor tissue [24]. Three significant radiological parameters, including the radiological tumor size, margin, and presence in area 3, showed a diagnostic performance comparable to that of the radiologists’ impression. Thus, these parameters would be reliable to predict FNI in the clinician’s viewpoint and to decrease interobserver reliability.

This study had certain limitations. Patients with preoperative facial weakness were excluded from this study. Preoperative facial weakness does not always lead to FNI in incomplete facial paralysis [8]. Neural invasion by cancer cells or compressive local ischemia or inflammation by a growing tumor may cause facial weakness [14,25] Surgeons usually consider facial reanimation surgery for patients with preoperative facial weakness. This study aimed to help surgeons prepare for unexpected facial nerve injury and consult patients before surgery. 

Therefore, patients with preoperative facial weakness were excluded from this study. In addition, histological FNI was confirmed using frozen biopsy to secure adequate resection margin and to avoid residual tumors. MRI is superior to CT in evaluating the extent of parotid cancer, and CT is usually used to evaluate the lymph node metastasis in parotid cancer. As we enrolled the patients with incidental parotid cancer, there were some patients lacking preoperative MRI. This study was not aimed to evaluate the differences or superiority between MRI and CT. 

Tumor location and size, which were radiological factors, can be assessed by CT. The radiologists’ impression was difficult to define, and conflicting findings between different radiologists’ impressions and the three factors were not clearly summarized. This impression is intuitive and can vary among radiologists because an accumulation of experience is necessary to increase the diagnostic reliability. The detailed division of the gland into compartments would useful to establish a more objective predicting method for FNI. These diagnostic methods could compensate for each other and be beneficial to improve the diagnostic value of objective radiological parameters. 

## 5. Conclusions

The tumor size, speculated margin, and presence in area 3, which were found via CT or MRI, were predictive of FNI. Additional information, such as the histological grade, increased the diagnostic performance. Additional studies using both objective parameters and radiologist impressions should be conducted to predict FNI and help surgeons prepare for facial reanimation surgery more precisely. A more detailed division of the parotid gland would also help to analyze the risk of FNI in future studies.

## Figures and Tables

**Figure 1 cancers-14-01086-f001:**
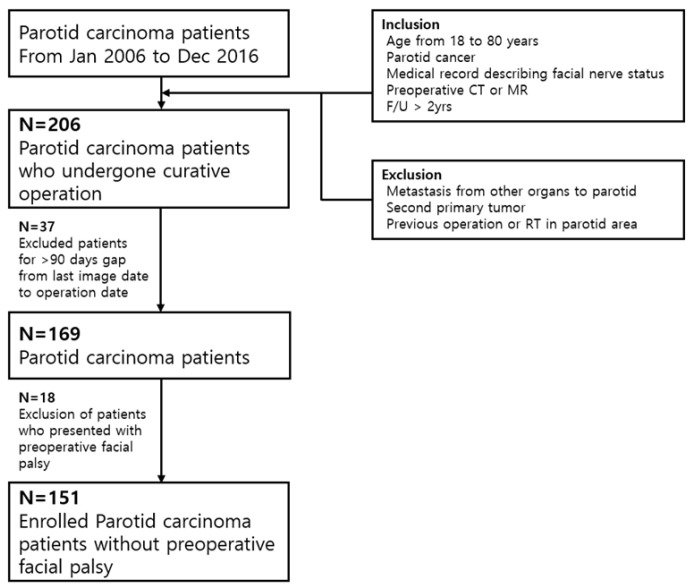
Enrolled patients.

**Figure 2 cancers-14-01086-f002:**
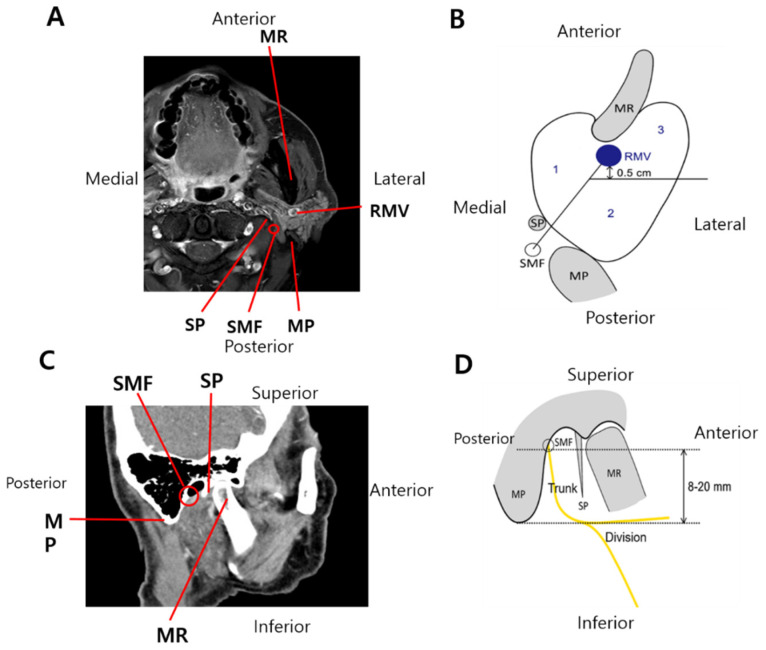
(**A**) T1-weighted magnetic resonance image in the axial plane, (**B**) schematic image of the partition of the right parotid gland. Two imaginary lines were drawn to divide the parotid gland into three compartments (area 1, area 2, and area 3), where the facial nerve trunk and division are likely to be located. (**C**) CT image in the sagittal plane, and (**D**) schematic image of the imaginary course of the main facial nerve trunk, which is considered to be located within 8–20 mm from the stylomastoid foramen. MR, mandible ramus; MP, mastoid process; RMV, retromandibular vein; and SMF, stylomastoid foramen.

**Figure 3 cancers-14-01086-f003:**
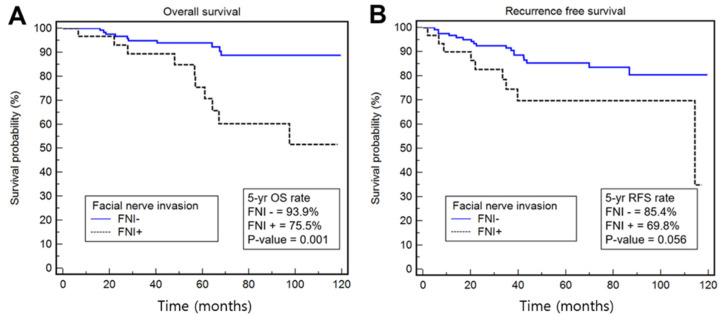
Survival outcomes. The overall survival rate (**A**) and recurrence-free survival rate (**B**).

**Figure 4 cancers-14-01086-f004:**
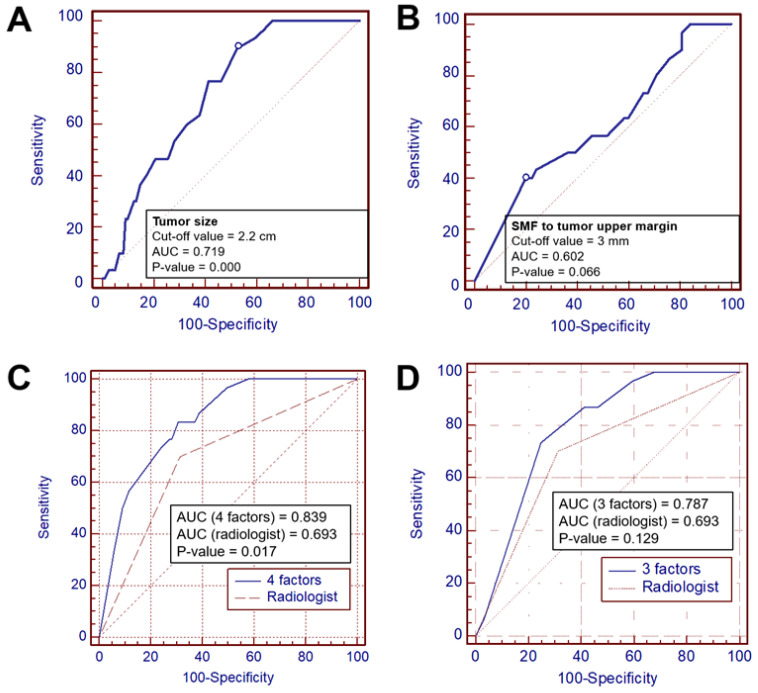
ROC curves for the optimal cutoff tumor size and distance from the stylomastoid foramen to the tumor (**A**,**B**) and for AUC estimation to distinguish histological facial nerve invasion using four objective factors and the radiologists’ subjective impression (**C**,**D**). SMF, stylomastoid foramen.

**Table 1 cancers-14-01086-t001:** Patient characteristics (*n* = 151).

Variable	Total (*n* = 151)	FNI (+) (*n* = 30)	FNI (−) (*n* = 121)	*p*-Value
Clinical factors				
Age (years), median (range)	55 (19–80)	56 (19–80)	55 (19–78)	0.630
Gender, male/female	73/78 (48.3/51.7)	21/9 (70.0/30.0)	52/69 (43.0/57.0)	0.008
Smoking history > 10 pack-years	32 (21.2)	6 (20.0)	26 (21.5)	0.858
Alcohol intake ≥ 1 drink/day	60 (39.7)	9 (30.0)	51 (42.1)	0.224
Charlson’s comorbidity index score ≥ 1	25 (16.6)	5 (16.7)	20 (16.5)	0.986
Histologic grade, low/intermediate/high	81/30/36 (53.6/19.9/23.8)	10/6/13 (33.3/20.0/43.3)	71/24/23 (58.7/19.8/19.0)	0.012
Pathological tumor size (mm), mean (SD)	24.1 (10.3)	31.2 (9.6)	22.4 (9.8)	0.000
cT category, T3–4	15 (10.0)	8 (26.7)	7 (5.8)	0.001
cN category, N1–3	24 (15.9)	10 (33.3)	14 (11.6)	0.004
Lymphovascular invasion	21 (13.9)	7 (23.0)	14 (11.6)	0.096
ECS (+)	56 (37.1)	18 (60.0)	38 (31.4)	0.004
Resection margin involvement	26 (17.2)	11 (36.7)	15 (12.4)	0.002
Follow-up duration, median (range)	62 (24–120)	62 (24–118)	61 (24–120)	0.767
Treatment				
Surgery alone	71 (47.0)	3 (10.0)	68 (56.2)	0.000
Surgery plus RT	72 (47.7)	26 (86.7)	46 (38.0)	
Surgery plus CCRT	8 (5.3)	1 (3.3)	7 (5.8)	
Last status				
NED	126 (83.4)	18 (60.0)	108 (89.3)	0.003
DOD/DOC	14/6 (9.3/4.0)	7/3 (23.3/10.0)	7/3 (5.8/2.5)	
AD	4 (2.6)	2 (6.7)	2 (1.7)	
Recurrence, any site	27 (17.9)	9 (30.0)	18 (14.9)	0.053

Data are expressed as *n* (%), unless otherwise specified. FNI, facial nerve invasion; SD, standard deviation; NED, no evidence of disease; ECS, extracapsular spread; RT, radiation therapy; CCRT, concurrent chemoradiation therapy; DOD: died of disease; DOC, died of another cause; and AD, alive with disease.

**Table 2 cancers-14-01086-t002:** Radiological factors distinguishing facial nerve invasion.

Variable	Total (*n* = 151)	FNI (+) (*n* = 30)	FNI (−) (*n* = 121)	*p*-Value
Level (location) of invasion division/trunk/both		24/4/2 (80.0/13.3/6.7)		
Margin, well-defined/spiculated/ill-defined	59/80/12 (39.1/53.0/7.9)	4/24/2 (13.3/80.0/6.7)	55/56/10 (45.4/46.2/8.3)	0.003
Tumor size (mm), mean (SD)	25.0 (9.5)	30.6 (7.5)	24.1 (9.5)	0.001
Presence in area 3	122 (80.8)	29 (96.7)	93 (76.9)	0.014
Tumor present at the main trunk course	34 (22.5)	10 (33.3)	24 (19.8)	0.113
RMV involvement, none/obliteration/displacement	114/27/9 (75.5/17.9/6.0)	18/9/2 (60.0/30.0/6.7)	96/18/7 (79.3/14.9/5.8)	0.023
SMF involvement (yes vs. no)	11 (7.3)	3 (10.0)	8 (6.6)	0.523
Distance from the SMF to the upper tumor margin (mm), mean (SD)	13.7 (11.4)	10.2 (9.5)	14.5 (11.7)	0.060
Positive impression of FNI by radiologists	59 (39.1)	21 (70.0)	38 (31.4)	0.000

Data are expressed as *n* (%), unless otherwise specified. SD, standard deviation; FNI, facial nerve invasion; RMV, retromandibular vein; and SMF, stylomastoid foramen.

**Table 3 cancers-14-01086-t003:** Multivariate analyses of factors predicting facial nerve invasion.

Variable	FNI	
	OR (95% CI)	*p*-Value ^a^
Margin, spiculated vs. ill–defined	4.35 (1.31–14.41)	0.016
Tumor size, 2.2 cm	6.02 (1.63–22.24)	0.007
Presence in area 3	8.98 (1.08–74.87)	0.042
High histological grade	3.65 (1.26–10.61)	0.017
RMV involvement (obliteration)		
RMV obliteration	1.54 (0.53–4.48)	0.426
RMV displacement	0.84 (0.13–5.41)	0.851
Positive impression of FNI by radiologists	3.55 (1.35–12.7)	0.012

FNI, facial nerve invasion; OR, odds ratio; CI, confidence interval; and RMV, retromandibular vein. ^a^ The multivariate models were subjected to a backward stepwise selection procedure with all variables with *p* < 0.05 in univariate results (Table 2), *p* < 0.05.

**Table 4 cancers-14-01086-t004:** Comparison of ROC curves for predictive radiological findings and the diagnosis by radiologists.

Variable	AUC	SE	95% CI	*p*-Value
Impression by radiologists	0.693	0.058	0.613–0.765	<0.001
Predictive variables				
Tumor size, margin, and presence in area 3	0.787	0.052	0.713–0.849	<0.001
Three factors + histological grade	0.839	0.047	0.770–0.893	<0.001
Difference between areas				
Three factors vs. radiologist	0.094	0.062	−0.028–0.216	0.129
Four factors vs. radiologist	0.146	0.061	0.027–0.265	0.017

SE, standard error; and CI, confidence interval.

**Table 5 cancers-14-01086-t005:** Diagnostic parameters for predictive radiological findings and the diagnosis by radiologists.

Variable	Sensitivity	Specificity	PPV	NPV	Accuracy
Impression by radiologists	70.00	68.60	35.59	90.22	68.87
Predictive variables					
Size, margin, and presence in area 3	76.67	72.73	41.07	92.63	73.51
Three factors + histological grade	83.33	69.42	40.32	94.38	72.19

## Data Availability

The data presented in this study are available within the article.

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
