# Peer review of "Preoperative Magnetic Resonance Image and Computerized Tomography Findings Predictive of Facial Nerve Invasion in Patients with Parotid Cancer without Preoperative Facial Weakness—A Retrospective Observational Study"

_cancers, 2022, doi:10.3390/cancers14041086_

Round 1
Reviewer 1 Report
- Title: should reflect the methodology: here retrospective observational study
- Title: should in general avoid abbreviations
- Abstract: "151 patients who had undergone tumor resection" unclear: all 151 patients without preop facial palsy? i.e. FNI (+) group (n = 30) means n =30 patients with CT/MRI signs if FNI but no facial weakness?
- Abstract: "anterolateral parotid region (area 3)," - means what: there is no international standard for different areas in the parotid.
- Abstract: "worse 5-year overall survival" - median follow-up has to be mentioned, otherwise validity of 5-year data cannot be considered by the reader
- Abstract: "significant factors in univariate analysis" - for what? FNI? or worse OS?
- Abstract: "significant factors in univariate analysis" - what was significant in multivariate analysis? for FNI? and for OS?
- Abstract: "speculate margin," - means what?
- Abstract. "Tumor margin" means what? R+ versus R- or a value in mm?
- Abstract: Tumor size: larger, I guees, but which size?
- Abstract: Basic data missing: Parotid cancer stages? Gender? Age?
- Introduction: "Tumors located in the superficial or deep parotid gland lobe have been identified using various radiological examinations" - delete, does not tell anything
- Introduction: "A multiplanar analysis of these landmarks" - be more precise, ref. 11 is only one retrospective trial. May be: seems to be ..
- Material: "biopsy-proven" - really? unusual? do you mean fine-needle CYTOLOGY? This is not a biopsy
- Figures:
Figure 1: N=37 patients excluded because of gap: see above, rule out selection bias
Figure 1: N=18 patients with preop facial palsy: OS same or different to FNI+ patients?
Figure 2: see above: validity/reliability of this area classification?
Figure 2: see above, differences between evaluation via CT vs MRI?
Figure 3: Typo months
Figure 3: p-value from log rank test? Not explained in main text/statistics
- Material: So ALL patients between 2006 and 2016 received fine needle, CT and MRI? If not: how many were excluded? why? Rule out selection bias by comparison of excluded and included patients
- Material: "Patients with an interval of more than 3 months" see above, how many? Selection bias
- Material: "FNI was defined as gross tumor infiltration confirmed by operators," - ? Gold standard is the confirmation by the pathologist!
- Material: "On axial CT 105 and/or MRI, the parotid gland was divided into areas 1, 2, and 3 or the medial, posterolateral, and anterolateral regions, respectively." - see above: Validation of this classification?
- Material: "with the overall radiologists' impression" - impression? means what in statistical terms?
- Material: Kaplan-Meier method not mentioned in statistics
- Results: "Preoperative CT were evaluated for 170 all patients, while MRI was used for 127 patients (84.1%)." - okay, see above. Effect on rules, only to have one method?
- Table 1: this is cTNM or pTNM, as all FNI+ patients should be pT4!
- Table 1: again resection margin: means what here? R+? < 5mm?
- Table 1: ECS is ECS+?
- Table 1: Histology type missing
- Table 1: why CCRT: chemotherapy is no standard for parotid cancer!
- Results: see above: We talk only about patient without clinical facial palsy? These were excluded?
- Results: "p < 0.05)." in generell: present exact p-values
- Table 2: first/third column, not all numbers readable
- Table 3: "spiculated vs. ill–defined" - see above, means what?
- Table 3: "obliteration" means what?
- Table 3: "impression" means what? And: Blinded evaluation?
- Methods: "Two radiologists who have experienced" - blinded for final FNI status?
- Table 1: "Histologic grade, low/intermediate/high"- does not exist for all types of parotid cancer. Please explain how this was possible for all tumors
- Discussion: Several times blank missing before references in brackets.
- Discussion: "Surgeons usually consider facial reanimation surgery for patients with preoperative facial weakness." - if facial EMG is showing pathological activity even without facial weakness, this is also a sign for facial nerve infiltration
- Results. "The 5-year overall survival rate of all patients was 90.0% ..". was FNI an INDEPENDENT risk factor? Multivariate analysis is missing
Author Response
We appreciate your kind review.
We revised this manuscript as reviewers’ recommendation.
- Title: should reflect the methodology: here retrospective observational study
- We revised this
- Title: should in general avoid abbreviations
- We revised this.
- Abstract: "151 patients who had undergone tumor resection" unclear: all 151 patients without preop facial palsy? i.e. FNI (+) group (n = 30) means n =30 patients with CT/MRI signs if FNI but no facial weakness?
- We added this “without facial nerve paralysis before surgery”
- Abstract: "anterolateral parotid region (area 3)," - means what: there is no international standard for different areas in the parotid.
- You are right. In this study, we suggested a new concept of anatomical compartment easily detected in CT or MRI. Details was described in method section.
- Abstract: "worse 5-year overall survival" - median follow-up has to be mentioned, otherwise validity of 5-year data cannot be considered by the reader
- We also added median follow-up period in abstract, which was described in results (62 months (range: 24–120 months)).
- Abstract: "significant factors in univariate analysis" - for what? FNI? or worse OS?
- We clarified this.
- Abstract: "significant factors in univariate analysis" - what was significant in multivariate analysis? for FNI? and for OS?
- “Tumor margin, tumor size (2.2 cm), and presence in area 3 were factors predicting FNI in the logistic regression model” They were significant factors related to FNI in multivariate analysis.
- Abstract: "speculate margin," - means what?
- In general, spiculated margin is characterized by lines of varying length and thickness radiating from the margins of the mass. It can be used in sonography, MRI, CT, and mammography.
- Abstract. "Tumor margin" means what? R+ versus R- or a value in mm?
- We apologize you for using obscure term. Tumor margin was different from margin involvement. We described margin shape and added “spiculated” before tumor margin to clarify the term.
- Abstract: Tumor size: larger, I guees, but which size?
- We added size, 2.2cm.
- Abstract: Basic data missing: Parotid cancer stages? Gender? Age?
- We agree with you, however, we eliminated the insignificant factors in abstract due to limitation of the number of words. Detailed factors were described in results.
- Introduction: "Tumors located in the superficial or deep parotid gland lobe have been identified using various radiological examinations" - delete, does not tell anything
- We deleted this.
- Introduction: "A multiplanar analysis of these landmarks" - be more precise, ref. 11 is only one retrospective trial. May be: seems to be ..
- We agree with you. With reference, we were to suggest our opinon to emphasizing the idea that, multiplanar analysis is better to evaluate the tumor status than single modality. We eliminated ref 11 at the end of the sentence.
- Material: "biopsy-proven" - really? unusual? do you mean fine-needle CYTOLOGY? This is not a biopsy
- We added “after operation”
- Figures:
Figure 1: N=37 patients excluded because of gap: see above, rule out selection bias
- These patients were excluded because of the possibility of discrepancy from initial radiological findings. All records had detailed descriptions of FNI available in the operation note, and the presence
Figure 1: N=18 patients with preop facial palsy: OS same or different to FNI+ patients?
- OS of the patients with preoperative facial palsy was worse than FNI (+) without facial paralysis, which was not significantly different.
Figure 2: see above: validity/reliability of this area classification?
- The area classification was suggested by two radiologists. With them, the classification was validated by two head and neck surgeons again. After that, we were able to use this classification in this article.
Figure 2: see above, differences between evaluation via CT vs MRI?
- There was no differences between two radiological modalities.
Figure 3: Typo months
- We corrected.
Figure 3: p-value from log rank test? Not explained in main text/statistics
- We revised and added this in method. The 5-year overall survival rate of enrolled patients was compared using Kaplan-Meier analysis and long-rank test.
- Material: So ALL patients between 2006 and 2016 received fine needle, CT and MRI? If not: how many were excluded? why? Rule out selection bias by comparison of excluded and included patients
- This is a main limitation of this study. Definitely, MRI is superior to CT in evaluating the extent of parotid cancer. CT was used to evaluate the lymph node metastasis in parotid cancer. Also, we enrolled the patients with incidental parotid cancer, who did not undergo MRI before operation. This study was not aimed to evaluate the differences or superiority between MRI and CT. Tumor location and size, which were radiological factors, can be assessed by CT. We just described the diagnostic process for parotid mass. Because this study was based on imaging studies were required to analyze this study. We can not compare the patients with imaging study with the patients without imaging studies. There was no patients who underwent surgery without preoperative fine needle aspiration biopsy.
- Material: "Patients with an interval of more than 3 months" see above, how many? Selection bias
- It can be selection bias. However, we do not use imaging findings more than 3 months before surgery to avoid discrepancy between imaging findings and operative findings and to establish operative plan more accurately.
- Material: "FNI was defined as gross tumor infiltration confirmed by operators," - ? Gold standard is the confirmation by the pathologist!
- We agree with you. The aim of this study was to establish the preoperative plan. Surgeons are advised to remove the involved facial nerve by parotid cancer. In operative field, FNI were evident in most cases and there was no way to confirm FNI pathologically on-site. Frozen biopsy to determine FNI on operation also needs resected nerve. This study is suggesting the way to increase the diagnostic concordance rate between radiological FNI and clinical/operative FNI, which help to plan facial reanimation surgery adequately.
- Material: "On axial CT 105 and/or MRI, the parotid gland was divided into areas 1, 2, and 3 or the medial, posterolateral, and anterolateral regions, respectively." - see above: Validation of this classification?
- The area classification was suggested by two radiologists. With them, the classification was validated by two head and neck surgeons again. After that, we were able to use this classification in this article.
- Material: "with the overall radiologists' impression" - impression? means what in statistical terms?
- Impression was based on tumor morphology (spiculated or obscure margin), tumor size, and invasion to adjacent structure, suggesting facial nerve invasion. Criteria of apparent diffusion coefficient (ADC) was also considered variable between radiologists. Tumor histology and grade were not considered. Tumor location, retromandivular vein invasion, and tumor grade were added to these factors. This was added in discussion section.
- Material: Kaplan-Meier method not mentioned in statistics
- We added this in method.
- Results: "Preoperative CT were evaluated for 170 all patients, while MRI was used for 127 patients (84.1%)." - okay, see above. Effect on rules, only to have one method?
- Incidental parotid cancer is confirmed after surgery. In these cases, initial imaging diagnostic modality would be CT without MRI. Thus, we wanted to find radiological findings and to evaluate them as predictive factors. The radiological findings indicating tumor location and tumor margin morphology, used in this study, can be evaluated by both CT and MRI.
- Table 1: this is cTNM or pTNM, as all FNI+ patients should be pT4!
- We agree with you. We revised this.
- Table 1: again resection margin: means what here? R+? < 5mm?
- We revised resection margin involvement. Positive resection margin is defined as carinoma or CIS at the resection margin as suggested by NCCN guideline.
- Table 1: ECS is ECS+?
- We revised this.
- Table 1: Histology type missing
- Histology types were so diverse. Instead, we categorized the histology type into low/intermediate/high grade and we described detailed histology in results section. The most common pathologic type of parotid carcinoma was mucoepidermoid carcinoma (n = 48, 29.6%), irrespective of the presence or absence of FNI. In patients without FNI, mucoepidermoid carcinoma (n = 38, 31.4%) was followed by carcinoma ex pleomorphic adenoma (n = 20, 16.5%). In patients with FNI, mucoepidermoid carcinoma (n = 10, 33.3%) was followed by adenoid cystic carcinoma, salivary ductal carcinoma, and carcinoma ex pleomorphic adenoma (n = 6, 20% for all). FNI was identified in 30 (19.9%) patients
- Table 1: why CCRT: chemotherapy is no standard for parotid cancer!
- We agree with you. RT is preferred for intermediate/high grade tumor, lymphovascualr invasion, lymph node metastasis, advanced T stage, and resection margin involvement. CCRT can also be recommended by NCCN guideline even though it is category 2B. CCRT is not absolute contraindication.
- Results: see above: We talk only about patient without clinical facial palsy? These were excluded?
- We excluded the patient with facial palsy before surgery. We can prepare facial reanimation surgery for the patients with facial palsy preoperatively even though all patients with facial palsy do not have FNI. Radiological findings predictive of FNI in patients without facial palsy would be meaningful more from the viewpoint of prediction.
- Results: "p < 0.05)." in generell: present exact p-values
- We revised these. Detailed p-values were described in tables. When estimated P-value is “0.00”, we usually use “p<0.001” instead of exact p-value.
- Table 2: first/third column, not all numbers readable
- We revised this.
- Table 3: "spiculated vs. ill–defined" - see above, means what?
- The marginswere described as well-circumscribed, indistinct (ill-defined), or spiculated (characterized by lines radiating from the mass). We added this in method section.
- Table 3: "obliteration" means what?
- Obliteration implies involvement. We revised this.
- Table 3: "impression" means what? And: Blinded evaluation?
- Impression was based on tumor morphology (spiculated or obscure margin), tumor size, and invasion to adjacent structure, suggesting facial nerve invasion. Criteria of apparent diffusion coefficient (ADC) was also considered variable between radiologists. Tumor histology and grade were not considered. Tumor location, retromandivular vein invasion, and tumor grade were added to these factors. This was added in discussion section.
- Methods: "Two radiologists who have experienced" - blinded for final FNI status?
- Yes, FNI status was blinded for two radiologists. In addition to objective measures, two experienced radiologists reviewed CT and/or MRI scans and determined the subjective impression of FNI Final FNI status was also blinded for them. We added this in method.
- Table 1: "Histologic grade, low/intermediate/high"- does not exist for all types of parotid cancer. Please explain how this was possible for all tumor.
- This grading system was based on WHO classification. Pathologists always try to describe histologic grade for salivary gland cancer in our hospital. Rarely, some salivary gland cancers were difficult to carteroize them. Pathologists concluded the grade based on the morphology, including invasiveness, number of mitotic figures/HPF, cystic component, border, mitoses, anaplasia, and perineural invasion among others. All grading schemes were somewhat cumbersome, intimidating and occasionally ambiguous, but evidence suggests that using a scheme consistently shows greater reproducibility than using our intuitive approach.
- Discussion: Several times blank missing before references in brackets.
- We revised this.
- Discussion: "Surgeons usually consider facial reanimation surgery for patients with preoperative facial weakness." - if facial EMG is showing pathological activity even without facial weakness, this is also a sign for facial nerve infiltration
- We agree with you. Facial EMG for the patient without facial nerve paralysis was not routine exam in our hospital. After study, we began checking facial EMG for the patients with suspicious findings from preoperative images.
- Results. "The 5-year overall survival rate of all patients was 90.0% ..". was FNI an INDEPENDENT risk factor? Multivariate analysis is missing
- FNI was not an independent risk factor because we did not perform multivariate analysis. Other study already presented that FNI is a significant prognostic factor. We just wanted to show that this study population followed those previous findings and to emphasize that the prediction of FNI has another pivotal role in the surgery of parotid cancer.
We appreciate your kind and detailed comment to improve this manuscript
Reviewer 2 Report
The Authors perfomed a retrospective evaluation of preoperative MRI and CT images to investigate radiological findings predictive of facial nerve invasion in patients affected with parotid cancer with no facial impairment before surgery. The manuscript is of interest, since its findings may help clinicians in predicting the involvement of the facial nerve on imaging and hence properly plan for surgical treatment.
- Please use gender instead of sex throughout the manuscript.
- How did you check for the hazard proportional assumption with the Cox regression model? Did you use the loglikelihood test? Wald test? Did you check for interaction amongst variables? Is the model adjusted for interaction.
- How did you check for multiple hypotesis testing?
Minor
Introduction
- Line 52. Correct typo. Neve resection.
- Line 57. US, CT, and MRI. Please define the acronyms.
M&M
- Line 98. Two radiologists who have experienced for 12 years (Y. J. C.) and 19 years (J. H. L.) Please rephrase.
Author Response
We appreciate your kind review.
We revised this manuscript as reviewers’ recommendation.
- Please use gender instead of sex throughout the manuscript.
=> We revised the term.
- How did you check for the hazard proportional assumption with the Cox regression model? Did you use the loglikelihood test? Wald test? Did you check for interaction amongst variables? Is the model adjusted for interaction. How did you check for multiple hypotesis testing?
=> Fundamental assumptions behind the method were that radiological factors found in CT or MRI would not be relevant and the probability of vilataion of this assumpition is low. Unfortunately, we did not check Shoenfeld residuals. We obtained unadjusted hazard ratios (HR) for all clinical factors (model 1) and then adjusted HR with progressive adjustments for stage (model 2) using log-rank test and logistic regression test in SPSS. The t-test for continuous variables and the χ2 or Fisher's exact test for categorical variables were performed to analyze significant factors affecting FNI, assuming a multivariate normal distribution follwed by individual planned hypothesis testing. The association between FNI and factors was analyzed. Subsequently, the effect of radiological and clinical findings on the likelihood of relationship was examined using multiple logistic regression. All of them are not time-dependent factors different from survival outcomes. The multivariate models were subjected to a backward stepwise selection procedure. Some radiological findings and tumor grade are highly suspicious finding related to FNI. Tumor location, size, and margin morphology are independent factors. Their interactions were not significant in my model and the factors were not reported as a relevant factors elsewhere. We just checked interaction among singificant variables, we confirmed the model. This would be another limitation of this study.
Minor
Introduction
Line 52. Correct typo. Neve resection.
=> We revised this.
Line 57. US, CT, and MRI. Please define the acronyms.
=> We revised them.
M&M
Line 98. Two radiologists who have experienced for 12 years (Y. J. C.) and 19 years (J. H. L.) Please rephrase.
=> We rephrased this sentence.
We appreciate your kind and detailed comment to improve this manuscript.
Reviewer 3 Report
Has involvement of the mastoid part of the facial nerve been considered ? This point is very important for surgical planning.
Please specify the following points:
1 How many cases have been made on 1.5 and 3T
2 What elements were used for "impression of radiologists"
3 Have all patients undergone total parotidectomy or are have any cases of superficial parotidectomy ?
4 Why was direct nerve visualization not sought in patients sudied on 3T ?
Since MRI is the technique of choice in the evaluation of the parotid tumors , it is useful to evaluate separately the performance and CT and MRI; this is missing
Considering the multivariate analysis the irregular margins expression of infiltrative growth and the dimensions appear obvious parameters ; i think the division of the gland into compartments is more useful.
Author Response
We appreciate your kind review.
We revised this manuscript as reviewers’ recommendation.
Has involvement of the mastoid part of the facial nerve been considered ? This point is very important for surgical planning.
- We agree with you. Some patients had an extensive mass around the stylomastoid foramen and all of them were accompanied by facial palsy. In this study, we excluded the patients with facial palsy before surgery. Thus, involvement of the mastoid portion of the facial nerve was not considered.
Please specify the following points:
1 How many cases have been made on 1.5 and 3T
- For head and neck cancer, we prefer 3T MR imaging and 1.5T MR scanner was usually used for parotid benign tumor. Among 127 patients, 22 patients underwent 1.5T MR examination. We evaluated the extent of tumor and lymph node metastasis rather than traced the extratemporal facial nerve. Currently, we tried to check parotid tumor using 3T MRI.
2 What elements were used for "impression of radiologists"
- Impression was based on tumor morphology (spiculated or obscure margin), tumor size, and invasion to adjacent structure, suggesting facial nerve invasion. Criteria of apparent diffusion coefficient (ADC) was also considered variable between radiologists. Tumor histology and grade were not considered. Tumor location, retromandibular vein invasion, and tumor grade were added to these factors. This was added in discussion section.
3 Have all patients undergone total parotidectomy or are have any cases of superficial parotidectomy ?
- Total parotidectomy would be an adequate surgical method for extensive parotid cancer such as facial nerve invasion whereas there is little evidence that extensive operations result in better outcomes (Cracchiolo JR, et al. Otolaryngo Clin North Am 2016;49:41-24). In this study, total or radical parotidectomy was performed with resecting the involved facial nerve. Partial or superficial parotidectomy was applied for accidental parotid cancer (proven cancer after operation), confined lesion, and low grade tumor. We added this method.
4 Why was direct nerve visualization not sought in patients studied on 3T ?
- We agree with you. MRI tractography has recently been developed to detect perineural spread of parotid cancers. After completing this study, we are now working on that. However, we still need experience and specific modification on 3T MRI to trace the facial nerve and to apply this to clinical field.
Since MRI is the technique of choice in the evaluation of the parotid tumors , it is useful to evaluate separately the performance and CT and MRI; this is missing
- This is a main limitation of this study. Definitely, MRI is superior to CT in evaluating the extent of parotid cancer. CT was used to evaluate the lymph node metastasis in parotid cancer. Also, we enrolled the patients with incidental parotid cancer, who did not undergo MRI before operation. This study was not aimed to evaluate the differences or superiority between MRI and CT. We just described the diagnostic process for parotid mass. Because this study was based on imaging studies were required to analyze this study. We can not compare the patients with imaging study with the patients without imaging studies. There was no patients who underwent surgery without preoperative fine needle aspiration biopsy. Tumor location and size, which were radiological factors, can be assessed by CT.
Considering the multivariate analysis the irregular margins expression of infiltrative growth and the dimensions appear obvious parameters ; i think the division of the gland into compartments is more useful.
- We agree with you. We are willing to analyze a more detailed compartment of the parotid to determine the risk of FNI in in future deep learning-based studies. We added your comment in discussion section.
We appreciate your kind and detailed comment to improve this manuscript.
Round 2
Reviewer 1 Report
1´. Many queries were answered.
2. Abstract: "Spiculated Tumor margin, tumor size (2.2cm)," - wrong upper case use and space missing after 2.2
3. Abstract. "FNI" (+) group - like this may be misleading as it was not based on histology. Also everywhere in the main text: make clear that is "radiological FNI" - this remains the main weakness and I do still not read anything about this most important limitation. The gold standard as ground truth is histology. Still I miss a discussion on this issue. Due to the current wording, Figure 4 is misleading. The reader might think that the radiological FNI definition is equal to histology-proven FNI.
4. Conclusion/Abstract: with CT or MRI? both equally usable?
5. Discussion: Limitations are still not adequately described: without CT/MRI excluded? How many ? Missing Figure 1. Plus N=37 with >90 gap. Still a comparison of included and excluded patients is missing. At least, this must be mentioned in the Discussion. Better: Perform a comparison
6. Methods "Final FNI status was also blinded for them." - what is the "final FNI status"? due to the rebuttal: "the impression of the surgeon"? Explain in the text.
7. Table 1: How many patients underwent radical parotidectomy in both groups? and MOST IMPORTANT how many patients had a histology proven FNI? If this method should guide the surgeon, this is the most important issue.
Author Response
We appreciate your detailed review comments to improve this manuscript.
We tried to follow the reviewer’s opinion to organize this manuscript well.
- Many queries were answered.
- Abstract: "Spiculated Tumor margin, tumor size (2.2cm)," - wrong upper case use and space missing after 2.2
=> We are sorry about typos. We revised this.
- Abstract. "FNI" (+) group - like this may be misleading as it was not based on histology. Also everywhere in the main text: make clear that is "radiological FNI" - this remains the main weakness and I do still not read anything about this most important limitation. The gold standard as ground truth is histology. Still I miss a discussion on this issue. Due to the current wording, Figure 4 is misleading. The reader might think that the radiological FNI definition is equal to histology-proven FNI.
=> We agree with you. Histological confirmation on FNI is pivotal. Thus, we added this in method section. “FNI was defined as both gross invasion observed by surgeons and histological confirmation.”
=> In this study, we tried to find predicting factors related to true FNI, which requires nerve resection and reanimation surgery on time. Resection of the facial nerve should be decided during parotidectomy if there is any gross invasion of tumor into the facial nerve. We revised ambiguous terms on FNI in Figure 4 and this in discussion. “In addition, histological FNI was confirmed using frozen biopsy to secure adequate resection margin and to avoid residual tumor.” Also, we used the term “impression by radiologists” instead of “radiological FNI” to avoid misleading.
- Conclusion/Abstract: with CT or MRI? both equally usable?
=> Either CT or MRI, which can measure tumor size, margin shape, and location, is beneficial to predict FNI. We did not analyze this separately. We revised this.
- Discussion: Limitations are still not adequately described: without CT/MRI excluded? How many ? Missing Figure 1. Plus N=37 with >90 gap. Still a comparison of included and excluded patients is missing. At least, this must be mentioned in the Discussion. Better: Perform a comparison
=> We agree with you. Among 206 patients who underwent parotid surgery for parotid cancers, 37 patients were excluded due to lack of CT or MRI within 90 days before surgery. Thus, we subtracted 37 patients from 206 patients to 169 patients. Exclusion of these patients may be another selection bias for this study. Considering tumor growth and morphological change during more than 90 days, however, the closer time interval between imaging studies and operation is, the more accurate imaging studies would be. If we enrolled all patients regardless of time interval between imaging studies and surgery, underestimated patients would be included in this study, which could be another selection bias. In addition, imaging studies right before surgery are usually recommended, especially, in case of cancer-related operation. Comparison between included and excluded patients according to the time interval between imaging studies and operation would be another future study to guide exact or acceptable time point to achieve successful parotid surgery. We added this in discussion section instead of comparison. We are very sorry about this. Please, consider our purpose of this study.
- Methods "Final FNI status was also blinded for them." - what is the "final FNI status"? due to the rebuttal: "the impression of the surgeon"? Explain in the text.
=> We revised this. Histological final FNI status was also blinded for radiologists.
- Table 1: How many patients underwent radical parotidectomy in both groups? and MOST IMPORTANT how many patients had a histology proven FNI? If this method should guide the surgeon, this is the most important issue.
=> We agree with you. We added the number of patients who underwent radical parotidectomy. Radical parotidectomy was performed in all patients with FNI while total parotidectomy and partial parotidectomy were performed in 85 (70.2%) and 36 (29.8%) patients. FNI (+) was confirmed in all patients either during surgery (frozen section) or after surgery (permanent section).
We appreciate your careful reviews again.
Reviewer 2 Report
I do not have any further comment.
Author Response
Thank you for your kind review.
Reviewer 3 Report
the changes made are enough to consider the job for publication
Author Response
Thank you for your kind review.